# Computational Workflow to Study the Diversity of Secondary Metabolites in Fourteen Different *Isatis* Species

**DOI:** 10.3390/cells11050907

**Published:** 2022-03-06

**Authors:** Doudou Huang, Chen Zhang, Junfeng Chen, Ying Xiao, Mingming Li, Lianna Sun, Shi Qiu, Wansheng Chen

**Affiliations:** 1Research and Development Center of Chinese Medicine Resources and Biotechnology, Institute of Chinese Materia Medica, Shanghai University of Traditional Chinese Medicine, Shanghai 201203, China; hdd890920@163.com (D.H.); chen870826@126.com (C.Z.); cjf12347831@foxmail.com (J.C.); xiaoyingtcm@shutcm.edu.cn (Y.X.); sssnmr@163.com (L.S.); 2Department of Pharmacy, Changzheng Hospital, Second Military Medical University, Shanghai 200433, China; limingming.email@foxmail.com

**Keywords:** untargeted metabolomics, high-resolution mass spectrometry, real features screening, chemical characterization, genus *Isatis*

## Abstract

The screening of real features among thousands of ions remains a great challenge in the study of metabolomics. In this research, a workflow designed based on the MetaboFR tool and “feature-rating” rule was developed to screen the real features in large-scale data analyses. Seventy-four reference standards were used to test the feasibility, with 83.21% of real features being obtained after MetaboFR processing. Moreover, the full workflow was applied for systematic characterization of 14 species of the genus *Isatis*, with the result that 87.72% of real features were retained and 69.19% of the in-source fragments were removed. To gain insights into metabolite diversity within this plant family, 1697 real features were tentatively identified, including lipids, phenylpropanoids, organic acids, indole derivatives, etc. Indole derivatives were demonstrated to be the best chemical markers with which to differentiate different species. The rare existence of indole derivatives in *Isatis cappadocica* (*cap*) and *Isatis cappadocica* subsp. Steveniana (*cap*S) indicates that the biosynthesis of indole derivatives could play a key role in driving the chemical diversity and evolution of genus *Isatis*. Our workflow provides the foundations for the exploration of real features in metabolomics, and has the potential to reveal the chemical composition and marker metabolites of secondary metabolites in plant fields.

## 1. Introduction

Metabolomics has been widely applied by a broad spectrum of researchers interested in defining the biological roles of biomarkers, novel metabolites, and new drug candidates, as well as for disease diagnosis [1,2,3]. Untargeted and targeted metabolomics are the two most common methodologies for comprehensive and targeted analyses to provide global or simple metabolic overviews [4,5]. For sample sets of complex mixtures for which little information is available, untargeted metabolomics can provide an unbiased discriminatory analysis for global metabolite detection, and may also yield insights into biochemical functions [6]. Advances in liquid chromatography coupled with mass spectrometry (LC-MS) have become integral metabolomics platforms, with each resulting signal commonly being referred to as a “feature”. On average, the representative untargeted approach is usually performed with a tremendous amount of MS features across tens to hundreds of samples, which makes the exploration of metabolite features even harder [7]. Until recently, numerous open-source tools have been developed to define features for the generation of peak tables, with the most common ones being XCMS, MZmine2, and MS-DIAL [8,9,10]. Although an enormous and diverse collection of MS data is critical for metabolomics studies, the interpretation of results remains a challenge owing to the ever-increasing complexity of feature attributions, which complicates the determination of real features in peak tables.

In untargeted metabolomics, the peak table obtained with a feature-detection tool contains all the data acquired from a metabolomics study [11], implying that the quality of the peak table may influence data interpretation. Variations in experimental conditions and multiple sample sets may lead to erroneous assignments of feature attributions in MS data in a peak table, including parental features, adducts, fragments, isotopes, dimers and contaminants [12,13,14]. Real features cover, e.g., parental features, while adducts represent features according to real metabolites and are derived from real sample sources. Parental features, including quasi-molecular ions, are presented as a proton adduct of [M + H]^+^ in positive-ion mode or [M − H]^−^ in the negative-ion mode [13]. Meanwhile, adducts are formed by the addition of a molecular ion to a metabolite within the ion source. Notably, each adduct may coexist and correlate with the parental feature or other adducts. Such correlations inform the discovery of parental features accurately and rapidly, in particular for metabolites that have no parental features in MS. In contrast, the existence of enormous amounts of in-source fragments, which are produced during a series of dissociation events as a consequence of weak bonds that are broken in metabolites in the ion source, complicates feature annotation owing to the inherent diversity of metabolites [12]. Among the technical advances of the last decade, certain novel tools have attempted to address this challenge by focusing on improving the grouping of fragments with parental features; these tools include CAMERA, RAMClust and MS-CleanR [15,16,17]. MS-CleanR, as the most recently developed tool, utilizes full-featured generic filters and feature-clustering functions, and produces a user-friendly peak table [17]. Unfortunately, any inaccuracies in the selection of parental features will yield a peak table of poor quality. Given these potential issues, it is vital that a rule be implemented for accurate screening and selecting parental features among the inherent abundance of potential features with the purpose of “one feature to one metabolite”.

Chemical diversity is generated through the evolution of biosynthesis pathways in plants [18]. Although mass spectrometry can yield more structural information, the recognition of real features among thousands of features derived from diverse metabolites is still a significant obstacle in plant metabolomics. Here, we have developed a workflow which contains a self-developed tool named MetaboFR, aiming to obtain high quality peak tables in metabolomics. We demonstrate the efficiency of our workflow on the 14 different species from genus *Isatis*. Genus *Isatis* is a widely distributed plant, comprising 80 species from the Middle East to the Mediterranean region [19]. In China, *Isatis indigotica* Fort is important in traditional Chinese medicine (TCM), and is used for the treatment of fever, flu, and inflammation [20]. Its derivatives played a key role as an antiviral medicine in the SARS outbreak in 2002. Previous studies revealed the main compounds in *Isatis indigotica*, including indole alkaloids, organic acids, flavonoids, lignans, nucleosides, etc. Indigo, as a principal indole alkaloid in *I*. *indigotica*, has long been used as a natural dye around the world [21]. In addition, *Isatis tinctoria*, better known as woad, is widely used as a source of indigo and has been used as a medicinal plant in Europe for centuries. Modern studies show that it has anti-inflammatory, antitumor, antimicrobial and antioxidant activities [19]. However, only *Isatis tinctoria* and *Isatis indigotica* have attracted much attention, and little is known about the chemistry of other species in this genus. By employing our workflow, we are able to provide comprehensive chemical annotations based on real features screening (Appendix A), which will give us metabolic insights into the chemical diversity and plant evolution of secondary metabolites in *Isatis* L. genus.

## 2. Materials and Methods

### 2.1. Materials and Chemicals

In June 2018, the following *Isatis* species were planted on our behalf by the private company (Zealquest, Shanghai, China): *Isatis indigotica* Fort. (*ind*), *Isatis buschiana* (*bus*), *Isatis cappadocica* (*cap*), *Isatis cappadocica* subsp. Steveniana (*cap*S), *Isatis costata* C. A. Mey. (*cos*C), *Isatis tinctoria* L.(*tin*), *Isatis tinctoria* L. var. *tinctoria* (*tin*V), *Isatis japonica* Miq. (*jap*), *Isatis lusitanica*L. (*lus*), *Isatis glauca* Aucherex Boiss. (*gla*A), Tetraploid *Isatis indigotica* Fort. (4n = 28), (4*ind*), *Isatis oblongata* DC. (*obl*), *Isatis violascens* Bunge. (*vio*), and *Isatis minima* Bunge. (*min*). Information regarding the source of the seeds (14 species) is shown in Appendix A. Leaves of these 14 species were harvested four months after planting. All voucher specimens were deposited in the Institute of Chinese Materia Medica, Shanghai University of Traditional Chinese Medicine (SHUTCM, Shanghai, China). The phenotypes of 14 species are presented in Appendix A.

Pure distilled water used for LC-MS analysis was purchased from Watsons Water (Watsons, Hong Kong, China). HPLC-grade methanol, acetonitrile, and formic acid were purchased from Fisher Scientific (Thermo, Waltham, MA, USA). Warfarin and analytical-grade ammonium acetate were obtained from Sigma-Aldrich (Sigma-Aldrich, Darmstadt, Germany).

### 2.2. Standard Solution Preparation

A test mixture containing 74 reference standards was selected for validation of our workflow, and was prepared at three levels of concentration, i.e., 10, 50, and 100 ug mL^−^^1^ in MeOH. Seventy-four reference standards were purchased from Sigma-Aldrich (Darmstadt, Germany, purity ≥ 98%); detailed information is shown in Appendix A.

The following 12 reference standards for the characterization of genus *Isatis* were purchased from Dalian Meilun Biotechnology (Meilun, Dalian, China) and prepared at 100 ug mL^−^^1^ in MeOH: isovitexin, vicenin-2, guanosine, indigo, indicant, indirubin, indoxyl, isatin, pheophorbide a, pinoresinol, pinoresinol-4-O-d-glucose, and matairesinol-4-O-d-glucose (purity ≥ 98%).

### 2.3. Sample Preparation

Leaves of *Isatis* species were freeze-dried and passed through a 40-mesh sieve. The extract method from our previous study [22] was applied with some modifications. Briefly, the ground leaves (20.0 mg) were extracted in 8.0 mL methanol containing warfarin (200 ng mL^−1^) as an internal reference. The mixture was vortexed for 30 s and sonicated for 30 min (40 kHz, 250 W). The extract was then centrifuged at 4 °C for 10 min at 20,000× *g*. A 3-μL aliquot was subjected to analysis with UHPLC-QTOF-MS/MS (QTOF, quadrupole-time-of-flight). Finally, a blank sample (methanol) and a quality-control (QC) sample (aliquot of all samples) were also prepared for the LC-MS analysis.

### 2.4. LC-MS Analysis

For metabolic data acquisition for 14 species in genus *Isatis*, 1290 UHPLC combined with 6530 QTOF-MS (Agilent, Santa Clara, CA, USA) system was used. The samples were separated through ACQUITY BEH C18 (2.1 × 100 mm, 1.7 μm) (Waters Technologies, Milford, MA, USA) with a column temperature of 35 °C. In this research, gradient elution was applied using 2 mM ammonium acetate in water (A) and acetonitrile (B) as the elution phase. The elution procedure was conducted as follows: 95–90% A from 0 to 2 min; 90–48% A from 2 min to 10 min; 48–25% A from 10 min to 15 min; 25–5% A from 15 min to 25 min; and 5% A sustained from 25 min to 35 min. Another 4 min 95% A was used for re-equilibration, and the flow rate was set as 0.3 mL min^−1^. In addition, QC samples were employed every 10 injections to monitor the system stability. The mass spectrometer parameters were set according to our previous study with positive-ion as the acquisition mode [23].

### 2.5. Data Processing

The data processing workflow, shown in Figure 1, was divided into three steps. In step 1, the AnalysisBaseFileConverter (https://www.reifycs.com/AbfConverter/, accessed on 16 January 2022) software was used for raw mass spectrum data extraction, and MS-DIAL version 4.24 (https://prime.psc.riken.jp/compms/msdial/main.html, accessed on 16 January 2022) was used for MS peak detection and alignment with the following parameters [10]: retention time range, 1–30 min (1–18 min for reference standard samples); retention time tolerance, 0.15 min; MS^1^ mass range, 100–1200 (100–1000 min for reference standard samples); MS^1^ tolerance, 0.01 Da; MS^2^ mass range, 0–1200 (0–1000 min for reference standard samples); MS^2^ tolerance, 0.02 Da; minimum peak height, 5000 amplitude (15,000 for reference standard samples). Other parameters, including mass slice width and MS/MS abundance cut off, were set according to our previous study [23]. An internal standard was used for peak heights normalization.

In step 2, MS-CleanR (https://github.com/eMetaboHUB/MS-CleanR, accessed on 16 January 2022) was used to process the peak tables extracted from MS-DIAL [17]. In the MS-CleanR processing, several parameters, including minimum blank ratio and maximum retention time tolerance, were set according to our previous study [24], in which the minimum Pearson correlation coefficient was 0.8, and α = 0.05 indicated statistical difference. The peak table was generated as “MS_peaks-clusters_final” in MS-CleanR and was used for further MetaboFR processing.

In step 3, the peak table “MS_peaks-clusters_final” was processed by MetaboFR based on the “feature-rating” (FR) rule. A tutorial of MetaboFR is included in the Appendix A. The mass tolerance of adduct flagging and fragment removal was set to 0.01 and 0.05 Da, respectively. For adduct flagging, six adduct groups were defined and imported for characterization of genus *Isatis*: [M + H]^+^ to [M + NH^4^]^+^, [M + H]^+^ to [M + Na]^+^, [M + H]^+^ to [M + K]^+^, [M + NH^4^]^+^ to [M + Na]^+^, [M + NH^4^]^+^ to [M + K]^+^ and [M + Na]^+^ to [M + K]^+^.

### 2.6. Statistical Analysis

The normalized peak tables processed by different processing approaches were imported into SIMCA-P 14.1 (Umetrics AB, Umea, Sweden). All data were scaled by unit variance scaling, and all variables were pareto-scaled before autofitting. The principal component analysis (PCA) model was applied to determine unsupervised pattern recognition by importing different peak table compositions.

### 2.7. Metabolite Identification

An in-house library, containing 269 compounds reported from the entire genus *Isatis*, was established in order to obtain precise chemical interpretation results. Information about the 269 compounds, including chemical names and formulas, is listed in Appendix A, and the structures are displayed in Appendix A. To identify unknown metabolites, we also drew upon major natural product databases, including KNApSAcK, PlantCyc, LipidMaps, NANPDB and UNPD. Conveniently, an in silico software, MS-FINDER (https://prime.psc.riken.jp/compms/msfinder/main.html, accessed on 16 January 2022), integrates these databases and provides each compound with Simplified Molecular Input Line Entry Specification (SMILES) for chemical classification with the Classyfire function (https://classyfire.wishartlab.com, accessed on 16 January 2022) [25,26].

## 3. Results and Discussion

### 3.1. The Workflow of “Feature-Rating” Rule and MetaboFR

The purpose of our workflow is to remove both the false features from nonsample sources and interfering metabolites in order to improve the quality of peak tables and identification results [27]. First, as shown in Figure 1, MS-DIAL was used as a tool for peak extraction, alignment and normalization to generate a peak table in an untargeted metabolomics analysis in Step 1 [10]. In Step 2, generic filters and feature-clustering from the MS-CleanR tool were used to remove the interference signals and implement feature-clustering based on an MS-DIAL peak character estimation algorithm and Pearson’s correlation [28]. Any aberrant features resulting from signals in the “blank” sample, as well as any other unusual mass defects, were removed by the generic filters, which were also applied to establish a relative standard deviation threshold among the sample classes for the purpose of eliminating metabolites in the same sample class that were unstable [17]. Step 3 involves a FR rule based on the screening of real features. The real features are defined as “top-rated features” (TRFs) and “second-rated features” (SRFs), based on differences in their MS attributions. TRFs are parental features that either correlated with corresponding adduct ions or do not have a corresponding quasi-molecular ion, but rather, only corresponding adduct ions. In contrast, SRFs are defined as parental features that only have isolated quasi-molecular ions. For TRF screening, orienting the correlations between adducts and quasi-molecular ions can increase the efficiency of parental-feature identification. Notably, some molecules have no adduct form, e.g., [M + H]^+^ or [M − H]^−^ in MS; in such cases, it is imperative to take advantage of adduct correlations to identify parental features. However, for SRF screening, in-source fragments pose the greatest challenge when distinguishing with SRFs because both have isolated quasi-molecular ions in MS, which confounds the ability to distinguish between the two. In a summary, locating adduct correlations and reducing the number of fragments are two core concepts in the FR rule when exploring real features among thousands of MS signals.

Here, an R package called MetaboFR was developed to simultaneously capture adduct correlations and reduce in-source fragments by embedding the results after MS-CleanR in step 3 in our workflow. Desirable adduct correlations can be imported into the R package. The flagging of adduct types is based on the mass difference among each feature cluster, and the mass tolerance (typically 0.01 Da) is also tunable by user (as detailed in the tutorial on MetaboFR). All the features flagged with adduct types are regarded as TRFs and are labeled in the last blank column in the peak table. In addition, the reduction of in-source fragments to explore SRFs is another function of MetaboFR. Among the peak table generated from MS-CleanR, if the feature shown in MS^1^ (Average.Mz after MS-CleanR processing) also exists in MS^2^ (MS.MS. spectrum after MS-CleanR processing), based on other features in each feature cluster, it will be regarded as an in-source fragment and will be removed from the peak table. The mass tolerance for fragment screening is tunable for users (typically 0.05 Da). After removal of the fragments, the retained features (except retained TRFs) are regarded as SRFs. Finally, only the most intense TRFs (TRFs most) in each adduct correlation are manually selected to the final peak table, which are applied, together with SRFs, for metabolomic analyses.

### 3.2. Validation of Workflow by Applying Reference Standards

To validate our process, we applied a mixture of 74 reference standards to benchmark our workflow in negative-ion mode (Figure 2). Detailed information on the reference standards is shown in Appendix A. Three levels of concentrations along with QC and blank samples were acquired and imported into MS-DIAL for peak detection. The obtained peak table, containing 248 featuress was processed by MS-CleanR to generate the clustered peak table, with the result of 96 feature clusters covering 203 features (see Figure 2C and Appendix A). MetaboFR was applied to directly process Appendix A; 131 features were retained after adduct flagging and fragment removal (Figure 2C and Appendix A).

To locate the TRFs by adduct flagging, 21 adduct correlations derived from seven adduct types were imported into MetaboFR with a mass tolerance setting of 0.01 Da (see Figure 2D). Unlike conventional adduct flagging, flagging based on the adduct correlations is able to label the adduct types even for features without quasi-molecular ions in MS. In the fragment screening step, 72 features were removed based on the fragment screening rule shown in Figure 2D, and 40 dehydrogenation features without adducts were explored as SRFs. It should be noted that the flagged adduct features were defined as TRFs and were not be screened as fragment candidates. As a result of MetaboFR tool processing, 68 features were flagged as adduct types; the accuracy of flagging was 88.23%, whereas only five adducts were omitted due to unreasonable feature clustering observed by manual inspection (Figure 2E and Appendix A). Among the total of 131 features, the existence of 49.62% adduct features revealed that adducts exhibit great values for the screening of parental features, especially for the MS behavior of five features with no visible quasi-molecular ions (Appendix A). In order to explain the tool performance in more detail, Figure 3A shows a series of MS features from a retention time 7.41 min after processing by MS-CleanR [17]. Three compounds, i.e., epmedin C, etoposide and prednisolone, were identified among twelve features, comprising ten adducts and two fragments, by manually annotating. Notably, all three compounds exhibited multiple adduct types along with an absence of quasi-molecular ions, indicated that flagging the adduct types through adduct correlations is essential and effective. After the application of MetaboFR, nine adduct types were labeled and were regarded as TRFs, while two fragments derived from epmedin C and prednisolone were removed due to the existence in the MS^2^ spectrum of corresponding parental features. Finally, the most intense of TRFs were selected to represent the parental features of three compounds for further analysis. As shown in Figure 3B, one compound named p-hydroxybenzaldehyde (*m*/*z* 121.0295) was not detectable, as it was recognized as a fragment from an interfering feature, i.e., *m*/*z* 179.0355. After processing using the MetaboFR tool, 109 real features according to 73 reference standards within the total of 131 features were retained, including 69 TRFs and 40 SRFs (Appendix A), which significantly increased the ratio of real features in the peak table compared with processed with only MS-DIAL or MS-CleanR (Figure 2C). After the final selection of the most intense TRFs, 73 parental features and 22 interferences were included in the final peak table with an accuracy rate of parental features of 76.84% (Figure 2E). A manual inspection of final result is described in Appendix A.

### 3.3. Application of Workflow on Multicomplex Samples

To evaluate the application of our workflow on multicomplex samples, we set up an experiment for the metabolic profiling of 14 different species in plants of genus *Isatis*. Key chromatographic conditions were optimized to improve both the sensitivity and resolution. The ACQUITY BEH C_18_ column in positive-ion mode was selected because it has greater column capacity and yields better resolution and peak distribution (see Appendix A). The use of 2 mM ammonium acetate as the aqueous phase increased the number of detected ions and suppressed signal noise (Appendix A). According to published information on metabolites, the accepted standard mixture for metabolomics studies contains two flavonoids, five indole-related derivatives, one nucleoside, one chlorophyll and three lignans [29,30,31]. Such a mixture was injected into LC-MS to determine the *m*/*z* values and adduct types (Appendix A). Isovitexin, pheophorbide a, isatin and guanosine have common adduct types in positive-ion mode, in terms of the isolated quasi-molecular ions or after correlating with the corresponding adduct ions. In particular, pinoresinol 4-O-d-glucose only had adduct ions but no quasi-molecular ions. Based on the adducts identified in the different metabolites, six adduct correlations derived from [M + NH_4_]^+^, [M + Na]^+^, and [M + K]^+^ were selected to enhance the accuracy of the annotating adducts.

A total of 80 pieces of raw data (i.e., 71 leaf samples from 14 *Isatis* species, 1 blank sample and 8 QC samples) were acquired and imported into the MS-DIAL to generate a peak table (4895 features). The raw data were uploaded to Metabolights (MTBL4254). After processing in MS-CleanR, a total of 823 features were defined as interferences, and were removed by generic filters, yielding a new peak table named “MS_peaks-clusters_final” that contained 4072 features, along with 1935 feature clusters (Appendix A). Subsequently, Appendix A was treated using MetaboFR, together with six imported adduct correlations. A feature table containing 3426 features covering 1085 adduct flags was obtained. Meanwhile, 646 features were screened as the in-source fragments and removed directly by MetaboFR, whereas the remaining 2341 features were explored and treated as SRFs. After the final selection, the most intense TRFs were retained and 2825 features were included in the final peak table (see Appendix A). In order to evaluate the performance of MetaboFR, a detailed manual annotation of each feature was carried out with the peak table after the implementation of the generic filters with the rules shown in Appendix A (Appendix A).

### 3.4. Detailed Annotation of Peak Tables after MetaboFR and FR Analysis of Genus Isatis

Among the 4072 features listed in Appendix A, 1105 TRFs, 2020 SRFs, 860 fragments and 87 interferences (unknown adducts, isotopes, etc., referred to as unrecognized interferences) were annotated following a manual inspection. It should be noted that the feature attributions of fragments were labeled according to the Pearson’s Correlation (≥0.8) to the corresponding quasi-molecular ions within a time range of 0.03 min (Appendix A). Figure 4 presents the entire evolution of the peak table; the corresponding features with different attributions are listed in bar charts by comparing with detailed manual annotations. It is clear that a large quantity of fragments persisted even after the generic filters had been applied. After implementing our workflow and comparing the quality of the peak table obtained from MS-DIAL/MS-CleanR, the accuracy rate of real features was improved to 87.72%, whereas the number of fragments decreased to 9.38% (Figure 5A,B). As shown in Figure 5C,D, the false annotations compared with the manual inspection were caused from the false recognition of adducts and fragments by metaboFR processing. In Figure 5E,F, the occurrences of false recognized and unrecognized adducts or fragments were due to unreasonable feature clustering and generic filtration. As shown in Figure 5E, feature *m*/*z* 436.0651 (RT: 4.24 min) and *m*/*z* 420.0889 (4.30 min) were incorrectly recognized as [M + K]^+^ and [M + Na]^+^ within the same feature cluster. For the manual inspection, feature *m*/*z* 420.0889 exhibited no correlation and a separable retention time (>0.03 min) with *m*/*z* 436.0651, which was correlated to another feature, i.e., *m*/*z* 691.1317, at the same retention time in this cluster, indicating that *m*/*z* 420.0889 was an in-source fragment of *m*/*z* 691.1317. Among the unrecognized adducts, Figure 5F shows that two features, i.e., *m*/*z* 381.1161 (RT: 4.12 min, sodium adduct) and *m*/*z* 397.0894 (4.11 min, potassium adduct), exhibited adduct correlations after peak detection. However, *m*/*z* 397.0894 was removed by the generic filter due to its unstable MS behavior among the sample classes, which was unable to flag both features by MetaboFR.

It was also noted that the quality of the imported peak table can greatly influence the results of a metabolomics study. The two- (2D) and three-dimensional (3D) PCA results by importing only validated real parental features are shown in Figure 6C, indicating that 14 *Isatis* species were unsupervised clustered into five groups. Similar clustering results were observed when the MetaboFR and FR rules were implemented, revealing nearly the same distribution when validated real features were imported (Figure 6B,C). Interestingly, only the most intense imported validated TRFs (495 features, Appendix A) yielded five clusters, revealing that TRFs with a relatively high degree of confidence can be applied to output the most relevant results for complicated datasets (Figure 6D). Figure 6A presents a totally different distribution of different species in our PCA analysis compared with the other three PCA results, indicating that the lower ratio of real features may give rise to incorrect interpretations in metabolomics studies.

### 3.5. Chemical Analysis of Genus Isatis

Of the 2515 parental features, 109 metabolites were identified in the in-house database. The identification rate for the most intense TRFs (495 features; see Appendix A) was 74.54%, and that for the SRFs was 65.74%. Consequently, 1697 metabolites were tentatively identified to give a comprehensive chemical interpretation of 14 *Isatis* species (see Appendix A). The major metabolites were lipids, phenylpropanoids, organic acids and indole derivatives (Figure 7A). Five groups were clustered based on our PCA analysis; the relative contents of major metabolites among the 14 species is presented in Figure 7A. Four species in group I had a greater proportion of indoles and their derivatives. In particular the *4ind* compared with natural diploid progenitor (*ind*) had a higher proportion of indoles, alkaloids and phenylpropanoids. Group II contained four species derived from Europe and exhibited abundant organic acids and carbohydrates. Compared with groups I and II, groups III and V had a higher proportion of phenylpropanoids and fewer indole derivatives. The chemical compositions of *cap* and *cap*S were similar to those of the two desert species of *vio* and *min*, which were dominated by lipids yet lacked indole derivatives. Based on our characterization, it was found that the indole-related metabolites, which contain at least one indole moiety, exhibit tremendous chemical diversity and interspecies differences. As the representative metabolites in *Brassicaceae*, indole-related metabolites exhibit multiple bio-activities and are widely applied as pharmacological molecules [32]. Furthermore, 120 diversified indole-related metabolites were selected (Appendix A); their relative contents among 14 species are shown as a heat-map in Figure 7B. The result clearly shows that few indole-related metabolites existed in group IV, especially for the bare accumulation in *cap* and *cap*S. Figure 7C indicates the relative content of nine metabolites involved in the indole-related biosynthesis pathways. Accordance semiquantitative results were obtained, indicating that only small amounts of these metabolites were present in group IV, and only rarely were any of these metabolites detected in *cap* and *cap*S, demonstrating that the differences in accumulation were derived from the biosynthetic capacities of indole-related metabolites in plants of genus *Isatis*. In order to measure the diversity of indole-related compounds, we evaluated the contents of indole in plants of genus *Isatis* (Appendix A), proving that the existence of interspecific variability mainly occurred via downstream indole-related biosynthesis pathways among different species. Therefore, based on metabolomics data mining from a large dataset, the indole-related metabolites were found to be important chemical markers for distinguishing individual species of the genus *Isatis.* In addition, *cap* and *capS* may be used as natural mutants to reveal indole related biosynthesis in plants of genus *Isatis*. To summarize, we expect that our approach will facilitate metabolomics studies involving massive datasets, and as such, will provide more abundant chemical information in various scientific fields. Also, the proposed method will help to find suitable plants for the exploration of key metabolites during natural product biosynthesis.

## 4. Conclusions

The systematic characterization of complex mixtures of metabolites remains a substantial challenge, especially for real feature screening. Therefore, combined with the workflow of real features screening and unbiased identification, we obtained as much structural information regarding detectable metabolites as possible. In our research, MetaboFR (Appendix A), based on the MS-DIAL/MS-CleanR suite, was developed to screen real features and a comprehensive workflow, from raw data to final annotated peak list, was provided. The utility of this workflow is demonstrated by the fact that by analyzing secondary metabolites in 14 species of genus *Isatis*, 87.72% of real features were retained and 69.19% of the in-source fragments were removed. After careful manual checking, 1697 MS features were tentatively identified. Moreover, indole derivatives, which have been shown to be the medicinal basis of the use of *Isatis indigotica* Fort. and *Isatis tinctoria* L. [22,33], were explored as chemical markers by comparing differences in metabolites among 14 different species. More importantly, indole derivatives were rarely found in *cap* and *cap*S, which may provide natural mutant plants for indole derivative biosynthesis. Significant differences in indole derivative biosynthesis also indicated the chemical evolution of indole derivatives in genus *Isatis*. To summarize, we expect that our approach will facilitate metabolomics studies with massive datasets and provide more abundant chemical information for natural product research and other scientific fields.

## Figures and Tables

**Figure 1 cells-11-00907-f001:**
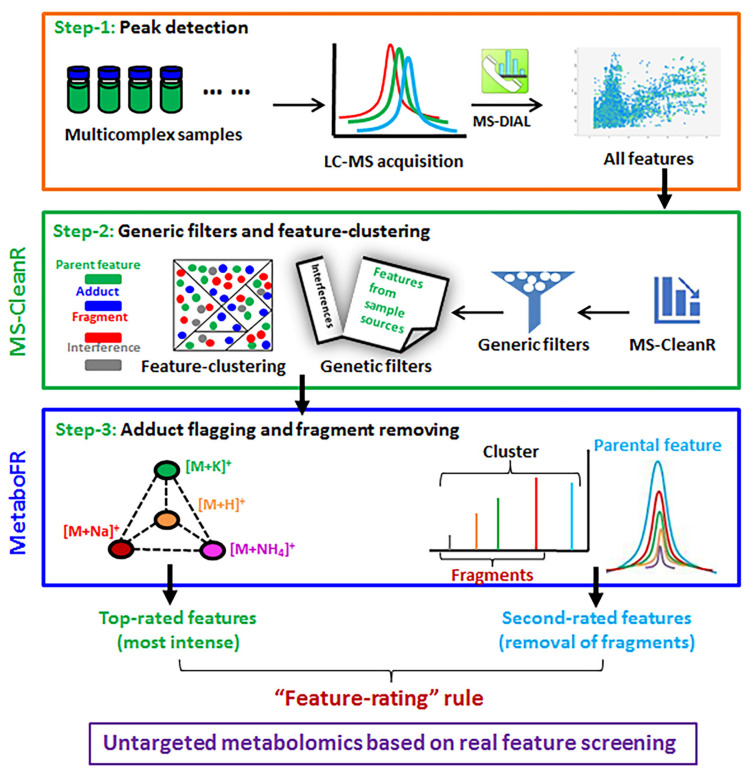
Data analysis workflow for comprehensive characterization of complex mixtures of metabolites based on the location of real features. Step 1: MS-DIAL was applied for the peak detection. Step 2: MS-CleanR was conducted utilizing the generic filters and feature clustering. Step 3: The developed MetaboFR tool combined with the “feature-rating” rule was applied for adduct flagging and in-source fragment removal.

**Figure 2 cells-11-00907-f002:**
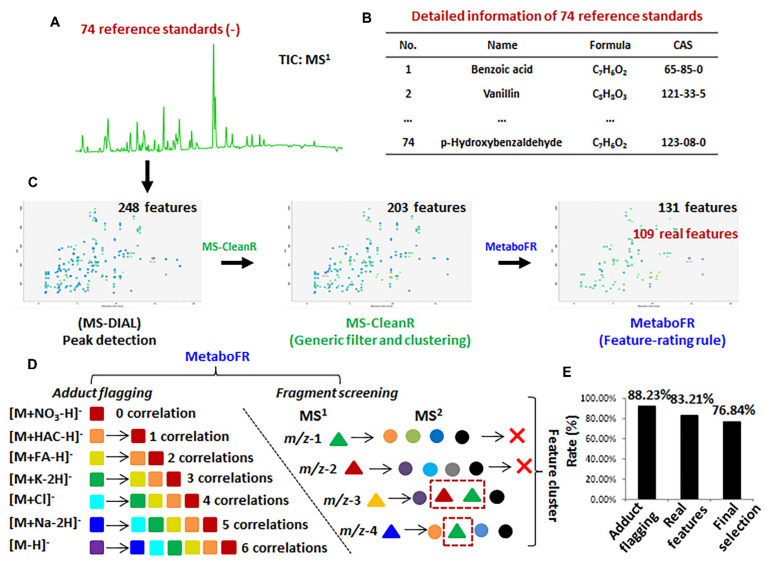
Seventy-four reference standards were used to validate our workflow. (**A**) A total ion current (TIC) chromatogram of 74 reference standards in negative−ion mode. (**B**) Detailed information about the 74 reference standards (Appendix A). (**C**) The evolution of feature amounts corresponding to three individual steps in our workflow. (**D**) The selection of adduct correlations and the rule of in−source fragment recognition. (**E**) The effect of our workflow on the final feature table of 74 reference standards.

**Figure 3 cells-11-00907-f003:**
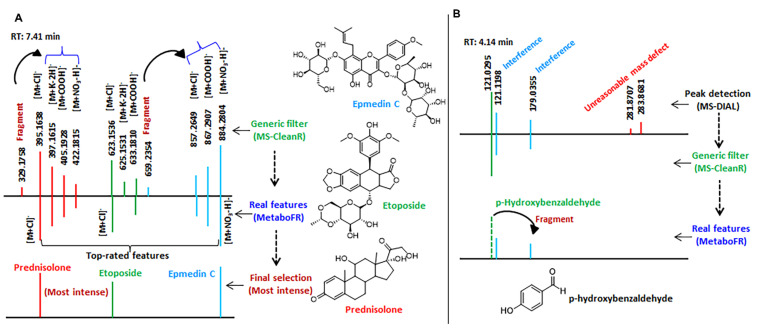
Two instances are illustrated to explain the performance of the tool in more detail. (**A**) The effect of MetaboFR and FR rule on the MS features at a retention time of around 7.41 min after processing by MS−CleanR. (**B**) One compound named p−hydroxybenzaldehyde (*m*/*z* 121.0295) could not be detected as it was recognized as a fragment from an interfering feature, i.e., *m*/*z* 179.0355.

**Figure 4 cells-11-00907-f004:**
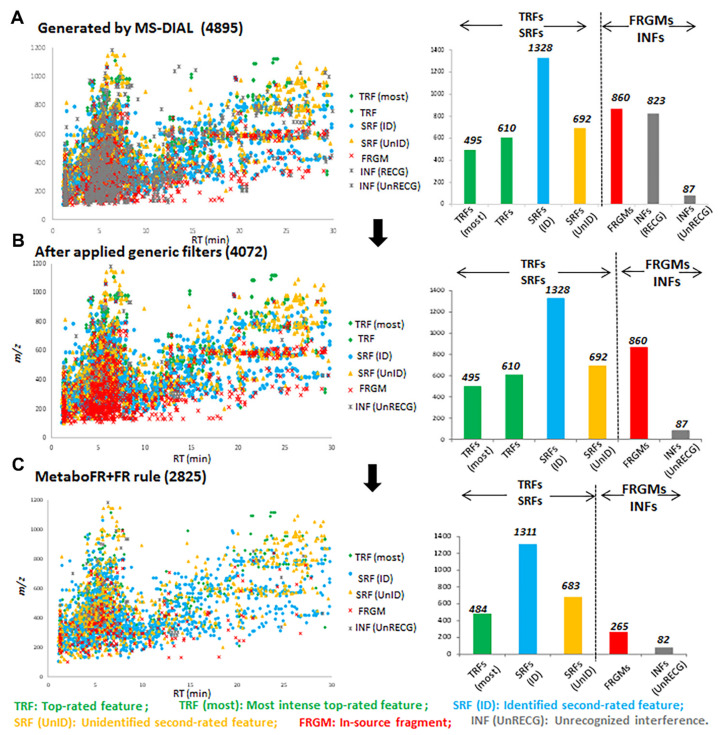
Evolution of the peak table by implementing the FR rule and quantifying the corresponding features with different attributions following a manual inspection, presented in bar charts. (**A**) The peak table generated by peak detection from MS−DIAL. (**B**) The application of generic filters eliminated the recognized interference. (**C**) MetaboFR combined with the FR rule preserved the most prominent TRFs and SRFs to yield a peak table that included most of the real features.

**Figure 5 cells-11-00907-f005:**
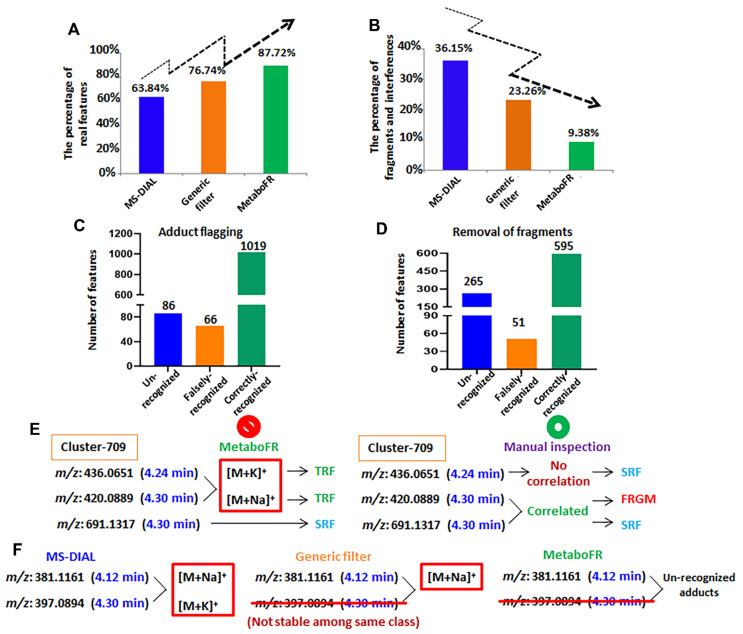
The effect of implementing our workflow to screen real features in metabolomics studies. (**A**) The percentage of real features identified by our workflow exceeds that of the co-implementation of MS−DIAL and MS−CleanR. (**B**) The existence of in-source fragments decreases substantially upon implementation of our workflow. (**C**) The effect of adduct flagging by MetaboFR tool. (**D**) The effect of fragment removal by MetaboFR. (**E**) False adduct recognition led to the false annotation of corresponding features by MetaboFR. (**F**) Some unstable features were removed by generic filters to cause the unrecognition of adduct correlations.

**Figure 6 cells-11-00907-f006:**
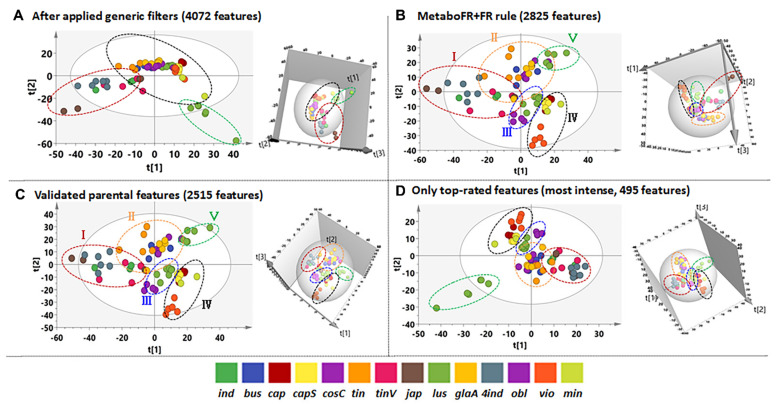
Different peak tables yield differences in the statistical results for metabolomics studies. (**A**) The 2D and 3D PCA results after the application of MS−CleanR (4072 features). (**B**) The 2D and 3D PCA results obtained by importing the features after the application of the MetaboFR and FR rules (2825 features) (**C**) The 2D and 3D PCA results obtained by importing only the validated parental features (2515 features) (**D**) The 2D and 3D PCA results obtained by importing only the most intense TRFs (495 features).

**Figure 7 cells-11-00907-f007:**
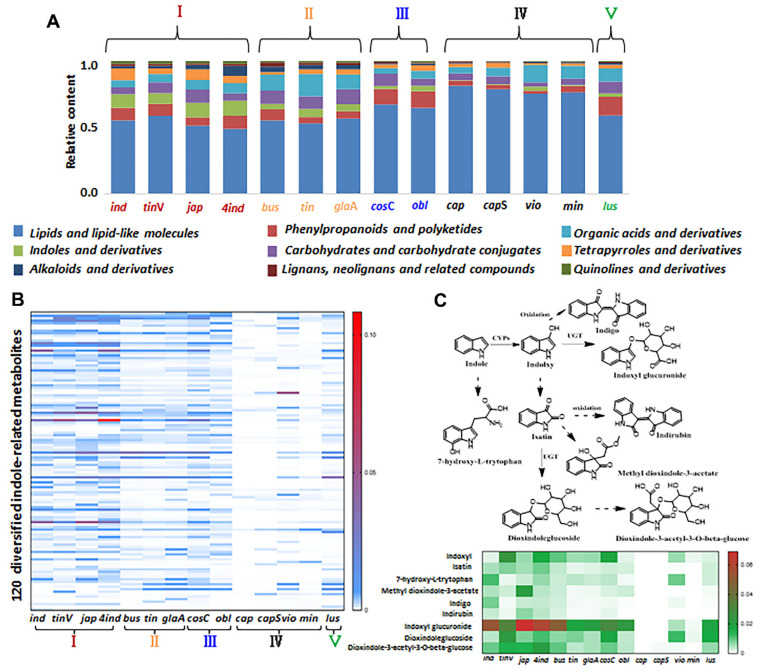
Comprehensive chemical interpretation and distribution of major metabolites among the 14 species of genus *Isatis*. (**A**) Bar chart showing the relative content of different classes of metabolites among the 14 species. (**B**) A heat-map showing the distribution differences of 120 indole-related metabolites among the 14 species within the five groups. (**C**) A heat−map showing the distribution differences of nine indole−related metabolites belonging to indole−related biosynthetic pathways among 14 species.

## Data Availability

Raw data were uploaded to Metabolights (MTBL4254).

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
