# Peer review of "Computational Workflow to Study the Diversity of Secondary Metabolites in Fourteen Different *Isatis* Species"

_cells, 2022, doi:10.3390/cells11050907_

Round 1

Reviewer 1 Report

In the manuscript by Huang et al. intitle “Computational workflow to study the diversity of secondary metabolites in fourteen different Isatis species”, the authors present the MetaboFR tool for the removal of features that are not related to a real molecule. They validated the workflow with a mixture of standards and applied it to the analysis of different Isatis species. Additionally, the use of an in-hose database of compounds allowed the identification of 109 metabolites from Isatis.

I believe that the article is consistent and should be published in Cells. However, I believe that the title and abstract are not matching the conclusion. If this is the first paper being published with the MetaboFR tool, the authors should give more importance to the tool in the title and abstract.

I addition, I have some suggestions and questions:

  • On my opinion, the authors should not abbreviate the species using “cap”, “capS”, but cappadocica, I. cappadocica subsp. Steveniana, etc. This is specially crucial in the abstract.

Pg. 4, L173: What the authors mean with “[M+H]+ to [M+K]+?

Pg. 5, Figure 1: In step 3, the authors states that fragments were removed. How the program can be sure that it is a fragment and not another molecule co-eluting?

Pg. 12, L382: The authors report that they annotated 1716 metabolites. Was the error calculated? Was the fragmentation checked for confirmation of the annotation?

Author Response

In the manuscript by Huang et al. in title “Computational workflow to study the diversity of secondary metabolites in fourteen different Isatis species”, the authors present the MetaboFR tool for the removal of features that are not related to a real molecule. They validated the workflow with a mixture of standards and applied it to the analysis of different Isatis species. Additionally, the use of an in-hose database of compounds allowed the identification of 109 metabolites from Isatis.

I believe that the article is consistent and should be published in Cells. However, I believe that the title and abstract are not matching the conclusion. If this is the first paper being published with the MetaboFR tool, the authors should give more importance to the tool in the title and abstract.

Response: Appreciate for your suggestion, we have rephrased the abstract section, and pointed the importance and necessity of MetaboFR. The attachment was our revised manuscript.

I addition, I have some suggestions and questions:

  • On my opinion, the authors should not abbreviate the species using “cap”, “capS”, but cappadocicaI. cappadocica subsp. Steveniana, etc. This is specially crucial in the abstract.

Response: Appreciate for your suggestion, we have revised “cap”, “capS” as “Isatis cappadocica (cap)” and “Isatis cappadocica subsp. Steveniana (capS)” in the abstract section.

Pg. 4, L173: What the authors mean with “[M+H]+ to [M+K]+?

Response: Appreciate for your question. “[M+H]+ to [M+K]+ means a parental feature both has [M+H]+ and [M+K]+ types at the same time. And the difference value between these two adduct types were applied to search both adduct types among the thousands of features.

Pg. 5, Figure 1: In step 3, the authors states that fragments were removed. How the program can be sure that it is a fragment and not another molecule co-eluting?

Response: Thanks for your great question. Recognition of fragment features and molecule co-eluting are both great challenges for the MS data analysis, especially for the complicated metabolomics data. In our research, the elimination of molecule co-eluting effect was mainly based on the m/z similarity at same retention window and consistency between different samples. If the fragment and co-eluted molecule both have high m/z similarity at same retention window and consistency between different samples, then we defined the co-eluted molecule as a fragment in our program. Of course, it is still difficult to discriminate the fragment and co-eluted molecule thoroughly. Our strategy can greatly simplify the screening of parental features, especially for the plant secondary metabolites identification.

Pg. 12, L382: The authors report that they annotated 1716 metabolites. Was the error calculated? Was the fragmentation checked for confirmation of the annotation?

Response: Thanks a lot for your great question. As we all known, metabolites annotation is a big bottleneck for metabolomics analysis due to the standards shortage and the existence of massive features in the MS. Thus, effective acquisition of real parental ion is the primary bottleneck to breakthrough to simplify the process of metabolomic analysis. However, some interfering ions, such as in-source fragments and contaminated ions, make it difficult to screen parent ions. In this research, MetaboFR was developed for screening of the real features. Of course, all the annotated metabolites were manually checked with calculated error and fragmentations based on MS-FINDER and in-house library, and we have created a new table (Table S10) containing the 495 top-rated features with corresponding mass error information, which shows that the error range is within 5 ppm for the majority of features. In MS-FINDER, the selection of annotation was according to the annotation score and database resources.

Reviewer 2 Report

Dear Authors,

The manuscript “Computational workflow to study the diversity of secondary metabolites in fourteen different Isatis species” develops an open-access tool, MetaboFr, for the purpose of obtaining the real peak table by embedding the results after MS-DIAL and MS-CleanR. The FR rule 433 was defined to convert complex a MS data matrix into a metabolite list as “one feature to one metabolite”.

It is well structured and contributes to facilitate metabolomics studies with massive datasets and provide more abundant chemical information for natural product research and other scientific fields.

I have no comments.

Author Response

The manuscript “Computational workflow to study the diversity of secondary metabolites in fourteen different Isatis species” develops an open-access tool, MetaboFr, for the purpose of obtaining the real peak table by embedding the results after MS-DIAL and MS-CleanR. The FR rule 433 was defined to convert complex a MS data matrix into a metabolite list as “one feature to one metabolite”.

It is well structured and contributes to facilitate metabolomics studies with massive datasets and provide more abundant chemical information for natural product research and other scientific fields.

I have no comments.

Response: Great appreciation for reviewing our manuscript.

Reviewer 3 Report

This manuscript contains a lot of data used and shows good way how to select important ones from a multitude. Also, the authors set up a computational workflow to study metabolites, to identify the origin of fragments obtained by mass spectrometry as well as the corresponding metabolites as markers of individual plants. Metabolomics is a complicated area of research, so any attempt to improve methods and techniques in this area is worth of attention. The results show that this idea can be acceptable and that further research should be done in that direction. Therefore, I consider that the manuscript should be published in the journal Cells without corrections.

Author Response

This manuscript contains a lot of data used and shows good way how to select important ones from a multitude. Also, the authors set up a computational workflow to study metabolites, to identify the origin of fragments obtained by mass spectrometry as well as the corresponding metabolites as markers of individual plants. Metabolomics is a complicated area of research, so any attempt to improve methods and techniques in this area is worth of attention. The results show that this idea can be acceptable and that further research should be done in that direction. Therefore, I consider that the manuscript should be published in the journal Cells without corrections.

Response:  Great appreciation for reviewing our manuscript.

Reviewer 4 Report

Dear Authors,

The present study describes a screening method for the study of metabolites of 14 Isatis species. The research subject is interesting and brings scientific important data in the field, as it deals with a subject that is currently of great interest. Some changes of the manuscript should nevertheless be performed in order to improve its quality. Following specific changes should thus be performed:

 Minor changes

         Line 17: chemical identification of Isatis?

Line 24: What do you mean by “a natural dye for a long time”?

        Please clarify abbreviations the first time they appear in the text. Please clarify the meaning of cap and capS.

        Line 37-38: genus Isatis is not a plant. Please rephrase all the sentence, it has no sense.

         Lines 39-44, 80-86: you have no citation.

Major changes

Abstract: Please include the main purpose of the study, it is not clear. Please clarify the part about indole derivatives. It is also not clear.

Introduction: this section should contain information regarding similar existing studies in literature and, in comparison, authors should emphasize the novelty and originality of their study. You have too much general information on methods of analysis (metabolomics, untargeted metabolomics) and too little on the species and compounds that specifically represent the subject of your study. The purpose of the study is not clear at all and needs to be rephrased. The last paragraph should clearly contain it. Also, you have little information about the genus that represents the subject of your study (why is in known, for which purposes/medicinal uses etc.). Please add further information and justifications and modify accordingly.

Materials and methods:

  • Please specify the exact source of the seeds that were used to specifically obtain the 14 species.
  • Please specify the 74 references that were used. It is too far to specify them only in Results
  • Are the methods described hereby completely new? If not and they were at least adapted, please add references.

Results and discussion: this part must contain comparison with similar existing studies in scientific literature in order to better help emphasizing novelty and originality of the present one. Please develop further. It is not enough to present these studies in the Introduction, you need to add correlations in this part, in order to add value for your manuscript, especially as you have made from discussions and results one section.  

Conclusions: Please offer potential perspectives for your study. Please insist on the compounds that are the most important in the composition of the tested species and may represent their biomarkers.

All these suggested changes should be performed in order to bring further improvements to the manuscript.

Author Response

The present study describes a screening method for the study of metabolites of 14 Isatis species. The research subject is interesting and brings scientific important data in the field, as it deals with a subject that is currently of great interest. Some changes of the manuscript should nevertheless be performed in order to improve its quality. Following specific changes should thus be performed:

Response: Thank you very much for providing valuable suggestion for our manuscript. The attachment was our revised version of manuscript according to your advice.  All the changes in the revised manuscript were labeled by the “tracked changes” function and our point-by-point responses to your comments are detailed below. 

 Minor changes

         Line 17: chemical identification of Isatis?

Response: Sincerely thank you for your careful inspection, we are sorry for inappropriate description, and we have revised the sentence as “…which may be useful for secondary metabolites annotation in plants of genus Isatis” in the revised manuscript.

Line 24: What do you mean by “a natural dye for a long time”?

Response: Thanks a lot for this question. In this research, the indole derivative of indigo is the main component in Isatis tinctoria and Isatis indigotica, which had used as natural dye for a long time, and we have corrected the sentence as “…a natural dye usage for a long time”.

        Please clarify abbreviations the first time they appear in the text. Please clarify the meaning of cap and capS.

Response: Thanks a lot for the problem you pointed out, we have revised “cap”, “capS” as “Isatis cappadocica (cap)” and “Isatis cappadocica subsp. Steveniana (capS)” in the abstract section.

        Line 37-38: genus Isatis is not a plant. Please rephrase all the sentence, it has no sense.

Response: Thanks a lot for the problem you pointed out, we have corrected as “plants in genus Isatis”.

         Lines 39-44, 80-86: you have no citation.

Response: Thanks for your suggestion; we have added relevant references in Lines 39-44, 80-86.

Major changes

Abstract: Please include the main purpose of the study, it is not clear. Please clarify the part about indole derivatives. It is also not clear.

Response: Thanks a lot for your valuable suggestion, the main purpose of our research was developing a new method for screening real feature of metabolomics analysis. In our research, we developed a new tool, MetaboFR, for screening real feature. Application of this tool, we demonstrated the compounds in 14 species of genus Isatis. Finally, among the thousands of features, we found the indole derivatives were screened as the key metabolites for differentiation of different species, especially for the extremely lower content in Isatis cappadocica (cap) and Isatis cappadocica subsp. Steveniana (capS). Thus, we have rearranged the abstract in the revised manuscript.

Introduction: this section should contain information regarding similar existing studies in literature and, in comparison, authors should emphasize the novelty and originality of their study. You have too much general information on methods of analysis (metabolomics, untargeted metabolomics) and too little on the species and compounds that specifically represent the subject of your study. The purpose of the study is not clear at all and needs to be rephrased. The last paragraph should clearly contain it. Also, you have little information about the genus that represents the subject of your study (why is in known, for which purposes/medicinal uses etc.). Please add further information and justifications and modify accordingly.

Response: Thanks a lot for your suggestion. Untargeted and targeted metabolomics are the two most common methodologies for comprehensive and targeted analysis to provide a global or simple metabolic overview. Until recently, numerous open-source tools have been developed to define features for the generation of peak tables, with the most common tools being XCMS, MZmine2, and MS-DIAL. Although an enormous and diverse collection of MS data is critical for metabolomics studies, the interpretation of results remains a challenge owing to the ever-increasing complexity of feature attributions, which complicates the determination of real features. Any inaccuracy in the selection of parental features will yield an incomplete list of features. Given these potential issues, it is vital that a rule be implemented for accurately screening and selecting parental features among the inherent abundance of potential features with the purpose of “one feature to one metabolite”. In our research, the main purpose of our research was developing a new method for screening real feature of metabolomics analysis. In China, Isatis indigotica Fort is treated as important herbal medicines in traditional Chinese medicine (TCM) used for the treatment of fever, flu, inflammation, and played a key role in SARS outbreak as an antiviral medicine in 2002. Previous studies revealed the main compounds in Isatis indigotica, including indole alkaloids, organic acids, flavonoids, lignans, nucleosides, and etc. Especially for indole alkaloids, researchers have reported multiple indole alkaloids in I. indigotica. Moreover, indigo, as a main indole alkaloid in I. indigotica, has been long used as a natural dye all around the world. In addition, Isatis tinctoria, which is well known as woad, is widely used as a source of indigo and medicinal plant in Europe for centuries, and the modern studies show it provides anti-inflammatory, anti-tumor, antimicrobial and antioxidant activities. However, only Isatis tinctoria and Isatis indigotica draw much attention, while other species were rarely studied. The analysis of chemical diversification and differentiation in Isatis L. genus will give us with metabolic insights to the medicinal material foundation for enhancing their medicinal value and application. Thus, we chose plants in genus Isatis for MetaboFR application. Please check the rearranged version in the revised manuscript.

Materials and methods:

  • Please specify the exact source of the seeds that were used to specifically obtain the 14 species.

Response: Appreciate for your valuable suggestion, the exact source of the 14 species seeds were provided in the supplementary material (Table S1).

  • Please specify the 74 references that were used. It is too far to specify them only in Results

Response: Thanks a lot for your suggestion, the information of 74 references, including name, formula, and CAS number were provided in in the supplementary material (Table S2).

  • Are the methods described hereby completely new? If not and they were at least adapted, please add references.
  • Response: Thanks a lot for your suggestion, MetaboFR is firstly developed in our research, based on the MS-DIAL/MS-CleanR suite. The relevant references were added in the revised manuscript.

Results and discussion: this part must contain comparison with similar existing studies in scientific literature in order to better help emphasizing novelty and originality of the present one. Please develop further. It is not enough to present these studies in the Introduction, you need to add correlations in this part, in order to add value for your manuscript, especially as you have made from discussions and results one section.  

Response: Appreciate for your suggestion. In this section, we mainly described the process of MetaboFR, including the application with references and genus Isatis samples. As we mentioned in the introduction section, MetaboFR was developed based on MS-DIAL and MS-CleanR. We have compared the performances between only processed with reported tools (MS-DIAL and MS-CleanR) and MetaboFR, the feature degeneracy has a great effect on the exploration of real features and removal of in-source fragments after MetaboFR processing. In this part, we compared the number of real ions by appling MS-DIAL, MS-CleanR and MetaboFR. After implementing the MetaboFR, the accuracy rate of real features in 14 species of genus Isatis increased from 63.84% to 87.72%, which greatly simplified the metabolites annotation. Please check the rearranged version in the revised manuscript.

Conclusions: Please offer potential perspectives for your study. Please insist on the compounds that are the most important in the composition of the tested species and may represent their biomarkers.

Response: Thanks a lot for your suggestion. In this section, we re-emphasized the purpose of this study and rearranged this paragraph. Please check the revised manuscript.

All these suggested changes should be performed in order to bring further improvements to the manuscript.

Round 2

Reviewer 4 Report

Dear Authors,

The present study describes a screening method for the study of metabolites of 14 Isatis species. The authors performed some of the changes suggested after the first round of review. Following specific changes should still be performed:

 Major changes

Abstract: Please check if it does not depass the recommended 200 words. In the actual form, the abstract still does not present a summary of your study, its purposes and does not follow its structure. Please re-organize and rephrase. Some explanations that you put in author’s response must be found in the abstract, as the main purpose is still not clear. Instead of the introductory phrase on metabolomics, authors should better put a phrase on the genus.

Introduction: I do not see the changes performed after the first round of review. It still does not contain information regarding similar existing studies in literature and the novelty and originality is still not clear. Please add all these. The purpose of the study is still not clear and needs to be rephrased. Also, you still have little information about the genus that represents the subject of your study (why is in known, for which purposes/medicinal uses etc.). Please modify accordingly.

Materials and methods: No references are found for the rest of the used methods in the manuscript, only for metabolite identification these references were added. Are they completely new (e.g. sample preparation, LC-MS analysis etc.)?

Results and discussion: If comparison with similar studies is not possible, then specify that your method and the obtained results are completely new.  If not, please compare with similar studies and state what you bring in novelty. Presentation and interpretation of results is important, but novelty also is.

All these suggested changes should be performed in order to bring further improvements to the manuscript.

Author Response

Dear editors:

Thank you so much for having our manuscript entitled “Computational workflow to study the diversity of secondary metabolites in fourteen different Isatis species” reviewed in a professional manner and for giving us more suggestions to revise the manuscript. All the changes in the revised manuscript were labeled by the “tracked changes” function and our point-by-point responses to the reviewers’ comments are detailed below.

Comments and Suggestions for Authors

Dear Authors,

The present study describes a screening method for the study of metabolites of 14 Isatis species. The authors performed some of the changes suggested after the first round of review. Following specific changes should still be performed:

 Major changes

Abstract: Please check if it does not depass the recommended 200 words. In the actual form, the abstract still does not present a summary of your study, its purposes and does not follow its structure. Please re-organize and rephrase. Some explanations that you put in author’s response must be found in the abstract, as the main purpose is still not clear. Instead of the introductory phrase on metabolomics, authors should better put a phrase on the genus.

Response: Thank a lot for your careful checking and great suggestion, we have rearranged this section according to your comments and limited the words within 200. The structure of our study mainly contains the establishment of our workflow for the real feature screening, testing the feasibility of our workflow with reference standards and the application of our workflow on the plant samples. By employing our workflow, it is able to improve the quality of peak table to make the metabolomics results more reliable, which is suitable for the comprehensive chemical interpretation of complicated plant samples and exploration of the key metabolites among thousands of features. Based on these, the sentences about the introductory phrase on metabolomics were condensed to explore more about our purpose in this study. Please check the revised version in revised manuscript. Thanks.

Introduction: I do not see the changes performed after the first round of review. It still does not contain information regarding similar existing studies in literature and the novelty and originality is still not clear. Please add all these. The purpose of the study is still not clear and needs to be rephrased. Also, you still have little information about the genus that represents the subject of your study (why is in known, for which purposes/medicinal uses etc.). Please modify accordingly.

Response: Appreciate for your great suggestion, we have rearranged this section to make the purpose of our study more clear. For the first paragraph, the main point is that the introduction of metabolomics and the recognition of the real features from massive MS information is still a great challenge. In the second paragraph, we have described that the composition of the peak table, and the importance to explore the real features from peak table to improve the quality of peak table in metabolomics. The final paragraph we have mentioned about the application of our workflow for the complicated plant samples, and the introductory information of genus Isatis. So the core concept of this study is to develop a workflow for the real features exploration in metabolomics, which is suitable for the chemical interpretation of massive data set, especially for the plant fields. Please check the revised version in revised manuscript. Thanks.

Materials and methods: No references are found for the rest of the used methods in the manuscript, only for metabolite identification these references were added. Are they completely new (e.g. sample preparation, LC-MS analysis etc.)?

Response: Thank a lot for your great question, the experimental methods in our study mainly contain sample preparation, LC-MS analysis, data processing, and identification. We have added corresponding references in the section of sample preparation, data processing and identification. For LC-MS analysis, the conditions of UPLC and MS was developed by ourselves and investigated based on the object of study. In order to achieve untargeted identification, the setting of parameters and selection of mobile phase was defined to suitable for as many components as possible in this study.

Results and discussion: If comparison with similar studies is not possible, then specify that your method and the obtained results are completely new.  If not, please compare with similar studies and state what you bring in novelty. Presentation and interpretation of results is important, but novelty also is.

Response: Thank a lot for your great suggestion. In this version, we have modified corresponding sentences to emphasize our results for comparing with the results processed without our workflow (routine method). By employing our workflow, the ratio of real features in peak table will increase significantly (Figure 4 and 5). Also, the PCA results have shown that the ratio of real features will influence the subsequent analysis (Figure 6). Please check the revised version in revised manuscript. Thanks.

All these suggested changes should be performed in order to bring further improvements to the manuscript.

We hope that the revised manuscript has addressed all the criticisms raised by the reviewers and that the manuscript is now suitable for publication in Cells. If there are any further questions, please let me know.

Sincerely Yours, 

Wansheng Chen